# The Senior Companion Program Plus: An Innovative Training Approach for Alzheimer’s Disease and Related Dementia

**DOI:** 10.3390/healthcare11131966

**Published:** 2023-07-07

**Authors:** Noelle L. Fields, Ling Xu, Ishan C. Williams, Joseph E. Gaugler, Daisha J. Cipher, Jessica Cassidy, Gretchen Feinhals

**Affiliations:** 1School of Social Work, University of Texas at Arlington, Arlington, TX 76019, USA; lingxu@uta.edu (L.X.); jessica.cassidy@uta.edu (J.C.); 2School of Nursing, University of Virginia, Charlottesville, VA 22903, USA; icw8t@virginia.edu; 3School of Public Health, University of Minnesota, Minneapolis, MN 55455, USA; gaug0015@umn.edu; 4College of Nursing and Health Innovation, University of Texas at Arlington, Arlington, TX 76019, USA; cipher@uta.edu; 5The Senior Source, Inc., Dallas, TX 75219, USA; gfeinhals@theseniorsource.org

**Keywords:** caregiving, lay provider, African American, culturally informed, training, Alzheimer’s disease, dementia

## Abstract

African Americans adults are disproportionately affected by Alzheimer’s disease and related dementias (ADRD) and are underrepresented in research about ADRD. Reducing gaps in the knowledge about ADRD in the African American community is important for addressing dementia care disparities. The existing psychoeducation interventions are often limited by cost and scalability; for these reasons, lay provider (i.e., volunteer) interventions are of increasing interest in ADRD research. The purpose of this study was to evaluate a training of African American Senior Companion (SC) volunteers (*n* = 11) with dementia-specific knowledge (i.e., Senior Companion Program/SCP Plus), as part of a culturally informed, in-home, psychoeducational intervention for African American ADRD family caregivers. Learning outcomes were measured pre- and post-training, using the Knowledge of Alzheimer’s Disease/dementia scale (KAD), the Sense of Competence Questionnaire, and the Preparedness for Caregiving Scale. The results showed significant improvements in knowledge of Alzheimer’s disease/dementia, one competence item, “It is clear to me how much care my care recipient needs”, and preparedness for caregiving. Overall, the study findings suggest the SCP Plus is a promising, culturally relevant, and potentially scalable lay provider training for ADRD with potential benefits that augment the existing Senior Companion Program.

## 1. Background

Alzheimer’s disease and related dementias (ADRD) affects over six million Americans and is the seventh leading cause of death in the United States [1]. Older African American adults are twice as likely to have ADRD as older white Americans, and among those aged 70 and older, 21.3% are living with ADRD [2]. Furthermore, ADRD disproportionately affects some racial groups such as African American populations, which is likely explained by health and socioeconomic disparities [3]. For example, using nationally representative data from the Health and Retirement Study, researchers found that racial differences in childhood adversity (social and economic) and socioeconomic status in adulthood placed African Americans at a higher risk of cognitive impairment [4]. Perceived discrimination may additionally affect cognitive functioning and in turn affect the risk of dementia in African American older adults [5]. Researchers using an epidemiologic cohort study of risk factors for cognitive impairment among older African Americans (Minority Aging Research Study) found that a higher level of perceived discrimination was associated with worse cognitive performance [6]. Similarly, in a study using data drawn from the Health and Retirement Study, greater everyday discrimination among Black participants was associated with a lower baseline memory and faster decline in memory [7]. In addition, older African American adults with ADRD may be more likely than non-Hispanic Whites to experience dementia care disparities related to access, cost, and receiving quality healthcare [8,9].

Gaps in the knowledge about ADRD in the African American community are additionally of continued concern [10,11]. Studies indicate African American adults may lack awareness about their increased risk of ADRD [12], and some have only basic knowledge of ADRD [13]. According to the Alzheimer’s Association [2], 55% of African Americans view the loss of memory/cognition as a normal part of aging rather than as a disease. Several studies report the knowledge of ADRD among African Americans is negatively impacted by stigma [14], myths about dementia [15], a mistrust of health care professionals [16], and systematic racism [17]. Key recommendations for improving ADRD-related outcomes have emphasized needs to increase education in under-served population groups, including African Americans [11].

Multicomponent psychosocial interventions have shown promise for an increased knowledge of ADRD. Research suggests that focal areas for interventions should include caregiver preparedness [18] and caregiver competency [19,20]. Interventions that enhance a sense of caregiver mastery are particularly relevant for African American ADRD caregivers as well [17]. Epps and colleagues implemented a nurse-led program that increased the knowledge of dementia among families in African American churches [21]. A pilot study of a person-centered, educational training designed for African American ADRD family caregivers led by a qualified trainer and attorney provided facts, community resources, legal information, and communication skills [22]. Results from this study showed no statistically significant changes in the knowledge of ADRD; however, themes from the qualitative interviews revealed an increase in medical knowledge about and factors contributing to ADRD [22]. While these studies demonstrate promise, they both rely on trained interventionists or professionals as a part of implementation. Trained interventionists/professionals are often costly, which may present challenges for scalability and cost effectiveness [23].

A largely unexplored area of research is the use of lay provider, peer-led interventions (i.e., using volunteers) to offer a more cost-effective approach for meeting the needs of ADRD families [23]. Research suggests volunteers can enhance non-pharmacological interventions for ADRD caregivers [24], and peers may be equipped to serve as interventionists for persons with ADRD and their family caregivers [25,26,27]. Moreover, the use of lay providers may better fit the cultural needs of participants as lay providers may have ‘insider knowledge’ that is important for establishing rapport, providing advice, giving feedback, and providing culturally meaningful support [28,29]. However, there is limited research related to the training aspect of volunteer-led interventions for persons with ADRD [30], and even less is known about culturally informed, lay provider interventions for African American volunteers and African American ADRD family caregivers.

To address these research gaps, the Senior Companion Program Plus (SCP Plus) was developed as a culturally informed, psychoeducational intervention for African American ADRD family caregivers that relies on the use of volunteer lay providers (i.e., Senior Companions) to improve the knowledge of ADRD, reduce caregiver burden, and improve caregiver coping. The SCP Plus utilizes the sociocultural stress and coping framework [31], which considers the unique cultural needs of ADRD caregivers. The SCP Plus augments the existing Senior Companion Program, which is a part of the national AmeriCorps Seniors in the United States.

The Senior Companion Program matches lower income, older adults to serve as Senior Companion (SC) volunteers to older adults who have one or more physical, emotional, or mental health difficulties and need support for daily living tasks to promote aging in place. SCs additionally provide caregiver respite and socialization for the older adult [32]. There is no income requirement to receive support from an SC. SCs typically provide help with light cooking and housekeeping, offer medication reminders, and accompany the client (i.e., person with ADRD) to social activities or medical appointments [32]. SCs receive an hourly stipend of USD 4.00 per hour for their time working with their clients [32].

While SCs serve clients with ADRD and their family caregivers, comprehensive training related to disease education and care for persons with ADRD are not included as a part of the Senior Companion Program [33]. However, SCs must attend a minimum of 20 h of pre-service orientation [3]. The pre-service orientation training may include topics such as the following: introductions, a welcome to AmeriCorps Seniors, team-building exercises, strategies for working with older adults, policies and procedures, Code of Conduct, mandatory reporting, elder advocacy, and services/support to help the volunteers with their own needs [3]. Each SCP site makes decisions about the content and structure of the pre-service orientation [3]. After the SCs’ first year of volunteering and for every year after, the SCP provides a minimum of 24 h of in-service training to the volunteers [3]. The in-services may take different forms, depending on the organization. In-service training topics may include helping older adults stay active and healthy, as well as coaching and problem-solving [3]. In-services are additionally a time to celebrate the volunteers’ accomplishments (e.g., number of years served, etc.). Each SCP site makes decisions about the content and structure of the in-services [3].

The purpose of this study was to examine the training of African American SC volunteers with dementia-specific knowledge as a part of the SCP Plus, a culturally informed psychoeducational program for African American ADRD family caregivers (for outcomes related to the family caregivers, see [34]). The aims of the SCP Plus volunteer training were to increase their knowledge of dementia, sense of competence, and preparedness for caregiving among the SCs (in order to prepare them to serve as ‘interventionists’ in the psychoeducational component of the SCP Plus). The SCs were already supporting ADRD family caregivers as a part of respite care; however, before the SCP Plus, they did not provide any other specific psychoeducational interventions for the ADRD family caregivers as their focus was on caring for the care recipients. Providing respite care and assistance for the person with ADRD is the typical focus of the SCs. We hypothesized that the SCs would significantly improve in their knowledge of dementia, sense of competence, and preparedness for caregiving following SCP Plus training.

The SCP Plus training relied on the best practices in designing training programs for older adults and is based on five principles of instructional design: “(1) learning is promoted when learners are engaged in solving real-world problems; (2) learning is promoted when existing knowledge is activated as a foundation for new knowledge; (3) learning is promoted when new knowledge is demonstrated to the learner; (4) learning is promoted when new knowledge is applied by the learner; and (5) learning is promoted when new knowledge is integrated into the learner’s world” [35] (pp. 44–45). See Figure 1 for an illustration of these principles applied to the SCP Plus training.

## 2. Methods

### 2.1. Participants

The study participants were SCs and recruited from SCPs in three sites in Texas, Louisiana, and Arkansas. Due to the challenges of COVID-19 at the time of this study and the participants representing a high-risk group, in-person data collection was halted 12 months into the planned 27-month recruitment period. Additionally, SCs were not allowed to meet with their clients due to the risk of infection, and the IRB (Institutional Review Board) did not allow any in-person data collection. At this time, 20 SCs had been recruited, with 11 of them randomly assigned to the SCP Plus intervention group to complete the SCP Plus training. SCs had to be the following: (1) currently participating in the Senior Companion Program; (2) currently providing respite services to African American dementia family caregivers; and (3) self-identified as African American.

After the SCP directors at each site identified the SCs who met the inclusion criteria, participants were randomly selected. The research team then contacted each Senior Companion. Participants were offered the opportunity to participate in the study, guided through the informed consent process, administered a baseline survey via phone, and randomized into one of two groups. The first group of SCs received the SCP Plus training and then delivered the modules to paired dementia caregivers, whereas the second group served as the control group who did not receive the training and had no weekly contact with caregivers. SCs assigned to the control group continued to receive their regular services, which encompass offering respite care to the family caregiver and providing personal care, light housekeeping, and meal preparation for individuals with dementia.

An *a priori* power analysis conducted using G*Power 3.1.9 determined that a minimum of 114 subjects would be needed to address our primary research objective. This sample size estimation was based on a small effect size (*f* = 0.12), with an alpha level of 0.05, power of 0.80, and a two-tailed approach. However, as a result of the COVID-19 pandemic, the study was halted, resulting in the recruitment and completion of training for only 20 SC participants. Subsequently, a post-hoc power analysis was conducted based on our final N of 20, which revealed that any result reflecting a moderate effect or higher (*d*_z_ ≥ 0.68) would yield adequate power of 0.80, alpha = 0.05, two-tailed.

### 2.2. Training Contents

Following best practice guidelines to develop easily accessible manuals that can be used for later reference [36], the research team designed the SCP Plus training manual. The training manual was divided into three main sections: (1) the background of the SCP Plus and program overview; (2) information related to the cultural aspects of caregiving; and (3) nine modules covering topics related to ADRD (see Table 1 for training overview). The training manual used Times New Roman, 22-point font size, and as recommended by Czaja and colleagues [36], the manual was written in plain, non-technical language. A consultant with African American ADRD caregiving research expertise additionally reviewed the manual’s content and language prior to implementation. Each section included blank, lined pages for notetaking.

The SCP Plus included a 12-h, in-person training course with the SCs in the intervention group. The training was designed for a group format and took place over the course of two days at each participating site. Three training courses were conducted (Dallas, TX, USA in May 2019; Little Rock, AR, USA in September 2019; and Baton Rouge, LA, USA in January 2020). Using results from a prior pilot of the SCP Plus, two research team members led the training, which featured several types of teaching strategies including rapport building through informal conversations at breakfast and lunch, the use of a written manual, time for questions, and both didactic and hands-on participation (e.g., role play).

### 2.3. Measures

Baseline surveys were conducted after the SCs consented to participate, which included their demographic information as well as the key measurements as described below. A questionnaire containing the main outcomes was additionally given to SCs, immediately after the SCP training.

The *Knowledge of Alzheimer’s disease/dementia scale* (KAD) [37] was used to assess attitudes and beliefs regarding Alzheimer’s disease/dementia including the following: (1) the epidemiology and etiology of Alzheimer’s disease; (2) perceived effectiveness of different forms of currently available treatments; (3) beliefs regarding the perceived threat of Alzheimer’s disease for oneself; and (4) how respondents learned about the disease. Results have shown the good reliability and validity of this scale in assessing attitudes and beliefs regarding Alzheimer’s disease/dementia among different ethnic minority dementia caregivers [38]. If a participant answered each item correctly, 1 point was given. The potential sum scores for KAD ranged from 0 to 14.

*Sense of competence* was measured by the subscale of “satisfaction with one’s own performance as a caregiver” from the Sense of Competence Questionnaire (SCQ). The SCQ consisted of 12 items that asked SC participants how satisfied they are with their own performance as a caregiver. Each item was measured from 1 = *strongly disagree* to 5 = *strongly agree*. The SCQ has been developed and repeatedly used to measure the preparedness in caregivers of dementia patients, and it has proved to be reliable and valid for this population [39,40]. The potential range of sum scores for SCQ was 12–60.

*Preparedness for caregiving* was measured by the Preparedness for Caregiving Scale (CPS) [41], which is a caregiver self-rated instrument that consists of eight items that asks SCs how well prepared they believe they are for the multiple domains of caregiving. Responses are rated on a 5-point scale, with scores ranging from 1 (*not at all prepared*) to 5 (*very well prepared*). This scale has demonstrated moderate to high reliability [42,43] and validity [41]. The potential sum scores for CPS ranged from 8 to 40.

### 2.4. Data Analyses

Univariate descriptive analyses of demographic information were first conducted to better understand the SCs samples. Because of the small sample size, medians were reported instead of means and standard deviations. Non-parametric McNemar tests were conducted for each item in the KAD scale. Non-parametric Wilcoxon signed-rank tests were computed to assess the changes over time among the sum scores of the KAD scale, as well as the individual item and sum scores of both the SCQ and CPS scales after the training. The study alpha was set to 0.05. All analyses were conducted with SPSS 28 for Windows.

## 3. Results

### 3.1. Demographics of the Senior Companions

Table 2 reports the descriptive information of SCs who completed the training (*n* = 11). All the SCs were female, with a median age of 70 ranging from 58 to 78. Approximately half of the participants were divorced (45.5%) and had “high school graduate or below” education (45.5%). All SCs reported attending a religious service regularly, and almost half of them attended the service nearly every day (45.5%). The SCs had taken care of their current client with ADRD for a median of 4 years (*range* = 1–15) and provided a median of 30 h weekly for their clients (*range* = 6–40). The majority of the SCs felt “very confident” or “confident” in providing care to their clients with ADRD (72.8%). More than half of the participants reported “difficulty” (54.5%) and “very difficult” (45.5%) in paying for their basic needs (i.e., financial strain). In general, the SCs had moderate levels of self-rated health (*median* = 3.0, *range* = 2–4).

### 3.2. Changes in Knowledge of Alzheimer’s Disease/Dementia Scale Scores

Table 3 shows the scores of the KAD before and after the SCP Plus. The results showed (bottom line in Table 3) the sum scores of the KAD were significantly improved (*z* = −2.97, *p* = 0.003) for SC participants, with a median KAD score of 10 (range 5–11) at the pre-test and median KAD score of 12 at the post-test (range 10–14). All of the 11 SC participants showed positive change scores between the pre- and post- tests (not shown in Table 3).

When looking at the individual items of the KAD before and after the training, the results indicated 3 items were answered correctly by 91% of SC participants in both pre- and post-tests. The three items were the following: “Most people with AD live in nursing homes”, (item 2), “Scientists have discovered a gene that causes most types of AD” (item 5), and “There is no known cure for AD” (item 9). The participants did not show changes after the training for item 7 (“Drugs are available to prevent AD”). The participants slightly decreased in their correct responses to item 13 (“People with AD usually die within a year or 2 after developing the disease”). For 2 of the 14 items, statistically significant improvements in the percentage of correct responses in the post-test were observed: item 8, “AD is just one of many types of dementia” (*p* = 0.016) and item 12, “Significant loss of memory and mental ability, commonly known as senility, is a normal part of aging” (*p* = 0.031).

When only looking at the KAD items correctly answered after the training, the results in Table 3 additionally show the overall knowledge of Alzheimer’s disease was relatively high at the post-test. The entire sample answered 100% correct for 7 of the 14 items (items 1, 2, 5, 8, 9, 11, and 12). A total of 80% or more of the overall sample correctly answered 4 of the 14 items (items 4, 7, 10, and 14). The rates of the correct responses to additional items included 54.5% of SC participants (item 6, “Drugs are available to treat the symptoms of AD”), 63.6% of participants (item 13, “People with AD usually die within a year or 2 after developing the disease”), and 72.7% of participants (item 3, “The first signs of AD usually occur before age 60”).

### 3.3. Changes of Competence Scores

Table 4 shows the scores of competences of caregiving, before and after the SCP Plus. While the sum score of the 12 items of the competence scale and changes in the majority of the single items were not statistically significant, SC participants did significantly improve in 1 competence item, “It is clear to me how much care my care recipient needs” (*z* = −2.598, *p* = 0.009).

### 3.4. Changes of Preparedness Scores

The results in Table 5 show that five out of the eight items improved in their median scores of preparedness for caregiving, after the SCP Plus training. Statistically significant improvements in their preparedness for caregiving were observed in three items. These items were the following: “How well prepared do you think you are to take care of care recipient’s emotional needs?” (*z* = −2.121, *p* = 0.034), “How well prepared do you think you are to find out about and set up services for the care recipient?” (*z* = −2.392, *p* = 0.017), and “How well prepared do you think you are to get the help and information you need from the health care system?” (*z* = −2.428, *p* = 0.015). Table 5 (bottom line) additionally shows the sum score of the preparedness for caregiving was significantly improved (*z* = −2.806, *p* = 0.005) for SC participants, with a median preparedness score of 24 (range 21–39) and 31 (range 23–40) at pre- and post-tests, respectively.

## 4. Discussion

The purpose of this study was to examine the knowledge of dementia, sense of competence, and preparedness for caregiving among SC volunteers before and after attending SCP Plus training courses. The results suggest that the participants gained knowledge of ADRD and improved in their preparedness for caregiving in sum scores and some single items. The participants only increased their sense of competence on one item, which is related to knowing how much help the care recipient needs after the SCP Plus training.

All the participants answered correctly that ADRD is not a “normal part of aging” after the training. This finding from the SCP Plus training is promising, given that research suggests over half of African American adults believe that cognitive decline/memory loss is a normal part of aging rather than a disease [2]. In addition, participants in the SCP Plus significantly improved their knowledge that ADRD is just one of many types of dementia. An improved knowledge in response to these items is additionally promising, as research suggests some African American caregivers are not familiar with the symptoms and progression of ADRD [10]. Furthermore, increasing education on ADRD among African American populations is identified as a public health strategy towards improving their rates of timely diagnosis and reducing the stigma towards the disease [11].

Research suggests that caregiver preparedness is an important target for ADRD caregiver interventions [18]. The participants in the SCP Plus training had improved overall scores on preparedness for caregiving, as well as on items related to taking care of the care recipients’ emotional needs, services for the care recipient, and obtaining help/information from the healthcare system. Although SCs receive a minimum of 20 h pre-service orientation training before they can serve as a volunteer and must complete a minimum of 24 h in-service training [32], there is no comprehensive ADRD education as a part of the Senior Companion Program. The SCP Plus adds value, as it augments the Senior Companion Program by providing education and training focused on caregiver preparedness for persons with ADRD.

Our findings suggest that the SCs improved their scores on one sense of competence item related to knowing how much help the care recipient needs. Given that SCs are tasked with supporting many of the daily needs of the care recipients, including assisting with grooming, eating, and exercise, it is critical that SCs are able to handle complex care needs that progress over the course of ADRD. This is particularly the case when considering the increasingly complex medical and nursing tasks that family/friend caregivers are assuming during the course of dementia [3]. Although research is mixed, focusing on caregiver competency may be an important aspect of multi-component ADRD caregiver interventions [19,20], and enhancing a sense of mastery (e.g., confidence and competence) is recommended for interventions with African American ADRD caregivers [17]. In this regard, the scope of the SCP Plus intervention closely aligns with key recommendations from the broader dementia caregiving literature to empower African American caregivers in the level of care provided to their care recipients through increasing education and awareness of strategies for addressing disease-related challenges [11].

Overall, our study suggests the five principles of effective instruction [35] provided a successful framework for the SCP Plus training (refer to Figure 1). The SCP Plus training addressed gaps in the knowledge of ADRD and caregiving (*problem*), through the phases of *activation*, *demonstration*, *application*, and *integration*. Based on Merrill [35], the *activation phase* included the Senior Companions recalling/drawing upon past relevant experiences as volunteers as a foundation for the new knowledge provided in the SCP Plus training. The *demonstration phase* was accomplished through role playing after each module. The *application phase* was completed through the post-testing, which suggested gains in knowledge, competence, and preparedness. Finally, the *integration phase* was implemented as a part of the SCP Plus intervention when the SCs shared their learning with ADRD family caregivers following the training.

## 5. Limitations

There are several study limitations to consider when interpreting the findings of this study. Due to the unforeseen impacts of COVID-19 that suspended the study, the sample size was very small. A larger sample size in the future is needed to better evaluate the SCP Plus training. Second, the sample was additionally limited to African American Senior Companions in three southern US states. Thus, a more representative sample in other states is needed to capture regional differences. Third, the volunteers were all female. While data from Ameri Corps Seniors reports that nearly 90% of SCs nationally are female, future studies should work to recruit male SCs [44]. Fourth, the present study did not measure the post-test for participants in the control group. Future studies can modify the design for post-testing senior companions in the control group, so that a comparison may be conducted of participants in the SCP plus group and control group.

## 6. Conclusions

One of the broad goals of the SCP Plus training was to build upon the skills and knowledge of SCs, by providing dementia-specific education and to encourage the use of community-based services. Improving education about dementia through a lay provider program such as the SCP Plus may be particularly important for African Americans, given that many may be unaware of their higher risk of ADRD [12]. Moreover, a systematic review by Werner and colleagues [45] found that inadequate knowledge of dementia was cited as a barrier to help-seeking for dementia. Overall, our study findings bolster support for the use of the Senior Companion Program as a promising platform for deploying culturally relevant trainings and interventions to support African American volunteers and the African American ADRD families that they serve in their roles as SCs.

Informed by the sociocultural stress and coping model, the SCP Plus was developed to address the need for accessible and culturally informed, community-based interventions for African American ADRD caregivers. Developing culturally relevant and cost-effective interventions are critical [46]. Interventions that are culturally informed are important, as family caregiving among African American ADRD family caregivers is complex due to the familial structures and the social and historical contexts of caregiving that are fundamental to many caregivers’ experiences [47]. The SCP Plus offers a culturally congruent, lay provider model for ADRD caregiving that is potentially cost-effective, as it relies on volunteers through the existing Senior Corps program. To promote cost-effectiveness, future SCP Plus trainings may be led by SCP staff (e.g., volunteer coordinator; program director). Monthly in-service education for SCs is a typical part of every SCP Program, and the SCP Plus may offer program directors additional options for their regularly scheduled trainings. Finally, the Senior Companion Program is offered nationwide, suggesting the potential for scalability in future iterations of the SCP Plus.

## Figures and Tables

**Figure 1 healthcare-11-01966-f001:**
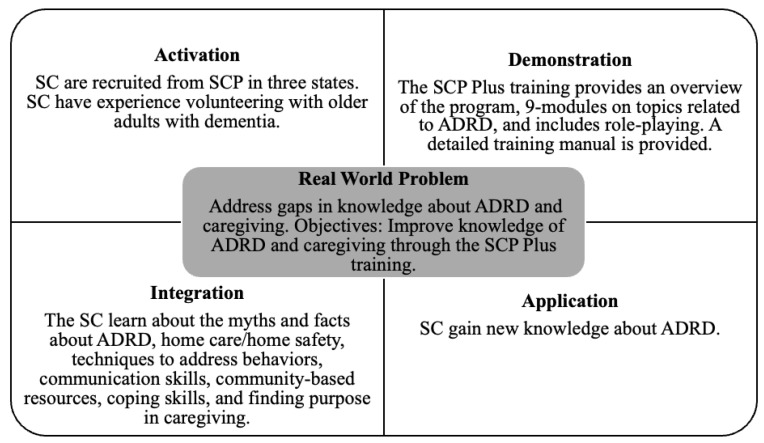
Principles of the Senior Companion Program Training (adapted from [35]). SC = Senior Companions; ADRD = Alzheimer’s diseases and related dementia; and SCP = Senior Companion Programs.

**Table 1 healthcare-11-01966-t001:** Overview of the Senior Companion Program Plus Training.

Topic	Content
Introduction	SCP Plus overview
	Cultural considerations of caregiving
Module 1	Facts about diagnosis and treatment of ADRD
	Role play
Module 2	Home care and home safety
	Role play
Module 3	Managing problematic behaviors
	Role play
Module 4	Communication with healthcare professionals
	Role play
Module 5	Communication skills to enhance interactions
	Role play
Module 6	Community-based programs
	Role play
Module 7	Coping skills
	Role play
Module 8	Coping with expectations
	Role play
Module 9	Finding purpose in caregiving
	Role play

**Table 2 healthcare-11-01966-t002:** Characteristics of Senior Companions in the Senior Companion Program Plus group at pre-test (*n* = 11).

Characteristics	Median	Range
	or *n* (%)	
Age	70	58–78
Women	11 (100.0%)	
Marital status		
Divorced	5 (45.5%)	
Never married	2 (18.2%)	
Married/cohabiting	2 (18.2%)	
Widowed	2 (18.2%)	
Education		
High school graduate or below	5 (45.5%)	
Some college	3 (27.3%)	
College graduation or above	3 (27.3%)	
Religious service attendance		
At least once a month	2 (18.2%)	
At least once a week	4 (36.4%)	
Nearly everyday	5 (45.5%)	
Length of care (years)	4	1–15
Daily care (hours)	30	6–40
Confidence		
A little bit confident	3 (27.3%)	
Confident	5 (45.5%)	
Very confident	3 (27.3%)	
Financial Strain		
Difficult	6 (54.5%)	
Very difficult	5 (45.5%)	
Self-rated health	3	2–4

**Table 3 healthcare-11-01966-t003:** Percentage of senior companions in the Senior Companion Program Plus group correctly answering true–false knowledge items about AD (*n* = 11).

	Knowledge of Alzheimer’s Disease Items	Pre-Test	Post-Test	*p*-Value
		*n* (%)	*n* (%)	
1	The primary symptom of Alzheimer’s Disease (AD) is memory loss (T)	9 (81.8)	11 (100)	*p* = 0.500
2	Most people with AD live in nursing homes (F)	10 (90.9)	11 (100)	*p* = 1.000
3	The first signs of AD usually occur before age 60 (F)	5 (45.5)	8 (72.7)	*p* = 0.250
4	Men are more likely to develop AD than women (F)	7 (63.6)	9 (81.8)	*p* = 0.500
5	Scientists have discovered a gene that causes most types of AD (F)	10 (90.9)	11 (100)	*p* = 1.000
6	Drugs are available to treat the symptoms of AD (T)	2 (18.2)	6 (54.5)	*p* = 0.219
7	Drugs are available to prevent AD (F)	9 (81.8)	9 (81.8)	*p* = 1.000
8	AD is just one of many types of dementia (T)	4 (36.4)	11 (100)	*p* = 0.016 *
9	There is no known cure for AD (T)	10 (90.9)	11 (100)	*p* = 1.000
10	AD can be diagnosed by a blood test (F)	8 (72.7)	10 (90.9)	*p* = 0.500
11	The number of people with AD is now higher than ever (T)	9 (81.8)	11 (100)	*p* = 0.500
12	Significant loss of memory and mental ability, commonly known as senility, is a normal part of aging (F)	5 (45.5)	11 (100)	*p* = 0.031 *
13	People with AD usually die within a year or 2 after developing the disease (F)	8 (72.7)	7 (63.6)	*p* = 1.000
14	AD is the most common type of chronic cognitive impairment among the aged (T)	8 (72.7)	9 (81.8)	*p* = 1.000
Sum scores of KAD (median and range)		10.00 (5–11)	12.00 (10–14)	*p* = 0.003

AD = Alzheimer’s disease, T = true, and F = false; the nonparametric McNemar tests were conducted for each item, and the nonparametric Wilcoxon signed-rank tests were conducted for the sum score. * *p* < 0.05.

**Table 4 healthcare-11-01966-t004:** Respondence of Senior Companions in the Senior Companion Program Plus group on sense of competence in providing care (*n* =11).

Sense of Competence Items	Pre-Test	Post-Test	Significance Test	*p*-Value
	Median	Median		
It is clear to me how much care my CR needs.	4	4	*z* = −2.598	*p* = 0.009 **
I am capable to care for my CR.	4	5	*z* = −1.00	*p* = 0.317
I feel that I do not do as much for my CR as I could or should.	3	2	*z* = −1.134	*p* = 0.257
I feel angry about my interactions with my CR.	4	4	*z* = −0.577	*p* = 0.564
I feel that in the past I have not done as much for my CR as I could have or should have.	3	3	*z* = 0.000	*p* = 1.000
I feel guilty about my interactions with my CR.	4	4	*z* = −0.577	*p* = 0.564
I feel nervous or depressed about my interactions with my CR.	5	4	*z* = −1.342	*p* = 0.180
My CR benefits from everything I do for him/her.	4	3	*z* = 0.000	*p* = 1.000
I feel pleased about my interactions with my CR.	4	5	*z* = 0.000	*p* = 1.000
I feel useful in my interactions with my CR.	4	4	*z* = −1.000	*p* = 0.317
I feel strained in my interactions with my CR.	3	3	*z* = −0.378	*p* = 0.705
I wish that my CR and I had a better relationship.	4	4	*z* = −0.447	*p* = 0.655
Sum scores of competences (median and range)	46.00 (36–55)	45.00 (37–55)	*z* = −0.673	*p* = 0.501

CR = care recipient; ** *p* < 0.01.

**Table 5 healthcare-11-01966-t005:** Respondence of Senior Companions in the Senior Companion Plus group on preparedness for caregiving (*n* =11).

	Preparedness for Caregiving Items	Pre-Test	Post-Test	Significance Test	*p*-Value
		Median	Median		
1	How well prepared do you think you are to take care of your CR’s physical needs?	3	3	*z* = −1.134	*p* = 0.257
2	How well prepared do you think you are to take care of his or her emotional needs?	3	4	*z* = −2.121	*p* = 0.034 *
3	How well prepared do you think you are to find out about and set up services for him or her?	3	4	*z* = −2.392	*p* = 0.017 *
4	How well prepared do you think you are for the stress of caregiving?	5	5	*z* = −1.300	*p* = 0.194
5	How well prepared do you think you are to make caregiving activities pleasant for both you and your CR?	3	4	*z* = −0.577	*p* = 0.564
6	How well prepared do you think you are to respond to and handle emergencies that involve him or her?	4	4	*z* = −0.816	*p* = 0.414
7	How well prepared do you think you are to get the help and information you need from the health care system?	2	4	*z* = −2.428	*p* = 0.015 *
8	Overall, how well prepared do you think you are to care for your CR?	3	4	*z* = 1.000	*p* = 0.317
Sum scores of competences (median and range)		24.00 (21–39)	31.00 (23–40)	*z* = −2.806	*p* = 0.005 **

CR = care recipient; * *p* < 0.05, ** *p* < 0.01.

## Data Availability

This research did not use a publicly available dataset.

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
