# Peer review of "The Senior Companion Program Plus: An Innovative Training Approach for Alzheimer’s Disease and Related Dementia"

_healthcare, 2023, doi:10.3390/healthcare11131966_

Round 1

Reviewer 1 Report

This study tackles an important topic and uses methods appropriate to achieve study aims. There are several points requiring clarification that would greatly improve the study.

1.       The introduction identifies the need to caregiver training via a volunteer network. The intervention in the study is at the volunteer (SC) level, but it’s unclear in the writing how this translates to the ultimate goal of supporting African American ADRD caregivers, particularly because the outcomes are primarily related to caregiving. That is, are the SCs going to be the lay volunteers that then train family caregivers or are they considered the caregivers themselves?

2.       Related to (1), if the point is that it is beneficial not to have to rely on professionals to train caregivers, who would be conducting the training of the SCs if this program were to scale to more sites? And would there be a costs associated with that training?

3.       I think a little more information about what type/amount of training the SCs receive should be included in the introduction. There is some info in the discussion, but I would move include more in the intro. Also, are the SCs matched with lower income persons in the community or are the SCs themselves supposed to be lower income? The text on page 2 sounds like it’s the latter, but I suspect the former is the correct interpretation.

4.       In describing the sample (and presenting results) it’s unclear how the control group was used, if at all. If not, why randomly select 11? If there was some data collection on the control group, it should be included in at least some of the tables.

5.       When describing the scales on page 5, the full range of potential scores should be included. I also suggest specifying that the questionnaires were given pre and post-training in the methods section. Related, what is the timing of when the questionnaires were administered (i.e. was it all in the same day/immediately post session)?

6.       In the results, Table 3 is frequently referred to as Table 2. Should the word “Medium” in the header of Table 4 be median?

Quality of English language is sufficient. There are some typos throughout that should be checked. Other points of clarification are requested in my comments above.

Reviewer 2 Report

This study investigated The Senior Companion Program Plus: An innovative training approach for Alzheimer’s disease and related dementia. The study, which findings suggest the SCP Plus is a promising, culturally relevant and potentially scalable lay provider training for ADRD with potential benefits that augment the existing AmeriCorps Seniors Senior Companion Program, is very interesting. 

Nevertheless, I have some comments and suggestions for authors, listed below: 

Authors should follow the instructions to authors requested by the journal.

-       In the text, reference numbers should be placed in square brackets [ ], and placed before the punctuation; for example [1], [1–3] or [1,3]. 

ð For example: Lines 33-34 “(National Institute on Agin, 2023)”, please note [1].

Abstract

-       The abstract exceeds 200 words (224 words). Authors should therefore delete the less important elements.

-       Lines 17-22: “The purpose of this study was to evaluate a two-day training of African American Senior Companion (SC) volunteers (N = 11) with dementia-specific knowledge (i.e., AmeriCorps Seniors Senior Companion Program /SCP Plus) as part of a culturally-informed, in-home psychoeducational intervention for African American ADRD family caregivers. . Learning outcomes were measured pre and post training using the Knowledge of Alzheimer’s disease/dementia scale (KAD), the Sense of Competence Questionnaire, and the Preparedness for Caregiving Scale.”

The letter N which indicates the number of people in the sample must be written in lower case. The authors should make this change throughout the manuscript. They scored two consecutive dots. They should delete one.

ð “The purpose of this study was to evaluate a two-day training of African American Senior Companion (SC) volunteers (n = 11) with dementia-specific knowledge (i.e., AmeriCorps Seniors Senior Companion Program /SCP Plus) as part of a culturally-informed, in-home psychoeducational intervention for African American ADRD family caregivers. Learning outcomes were measured pre and post training using the Knowledge of Alzheimer’s disease/dementia scale (KAD), the Sense of Competence Questionnaire, and the Preparedness for Caregiving Scale.”

Background

The authors explain that Alzheimer’s disease and related dementias disproportionately affects African American populations, which is likely explained by health and socioeconomic disparities (Alzheimer’s Association, 2023). These data could be developed further (in few lines), with specific studies. 

-       Line 71: The authors have separated author references by commas: “Bateman et al., 2016, Smith et al., 2018, Willis et al., 2018”. 

A semicolon should be used: 

ð “Bateman et al., 2016; Smith et al., 2018; Willis et al., 2018). 

     -    Lines 81-82 and Line 91: The authors forgot the comma: “i.e. Senior  Companions”; “(i.e. person with ADRD)”. The comma must be added.

=> “i.e., Senior Companions” 

=> “(i.e., person with ADRD)” 

-       Lines 97-100: " The purpose of this study is to examine the training of African American Senior Companion (SC) volunteers with dementia-specific knowledge as part of the SCP Plus, a culturally-informed psychoeducational program for African American ADRD family caregivers (for outcomes related to the family caregivers see Xu et al., 2023).”

The acronym SC having already been specified; the authors do not need to recall it in the sentence. So, the authors could write:

ð  " The purpose of this study is to examine the training of African American SC volunteers with dementia-specific knowledge as part of the SCP Plus, a culturally-informed psychoeducational program for African American ADRD family caregivers (for outcomes related to the family caregivers see Xu et al., 2023).”

-       Line 107: The reference of Merrill (2002), cited for the first time on line 107, is missing. The authors have to add it in the references. They should check that all references are listed in the References section.

ð Merrill, M. D. (2002). First principles of instruction. Educational technology research and development50, 43-59. https://doi.org/10.1007/BF02505024

-       Line 112: In Figure 1, several modifications are required. 

I appreciate that the authors have created a figure adapted from Merrill (2002). It could be better to put sections in order proposed in the Background section: “These include engaging the SCs in: 1) solving real world problems; 2) activating existing knowledge to support new knowledge;  3) demonstrating new knowledge; 4) applying new knowledge; 5) integrating new knowledge. ” (Lines 107-110)

In the section called “Integration” the authors could note “SC” instead of “Senior Companions”.

Similarly, in the section called “Activation”, the sentences “Senior Companions are recruited from Senior Companion Programs in three states. Senior Companions have experience volunteering with older adults with dementia.” could be replaced with: 

ð “SC are recruited from SCP in three states. SC have experience volunteering with older adults with dementia.”

In the section called “Problem”, the sentence: “Improve knowledge of ADRD and caregiving through the Senior Companion Program Plus (SCP Plus) training.” Should be replaced with:

ð “Improve knowledge of ADRD and caregiving through the SCP Plus training.”

Finally, in the section called “Application”, the authors should replace the sentence “Senior Companions gain new knowledge about Alzheimer’s disease and related dementia (ADRD) and increase competence and preparedness for caregiving.” with   

 => “SC gain new knowledge about ADRD and increase competence and preparedness for cargiving.”

The authors should add a legend below the Figure 1, specifying the meaning of the acronyms:

ð SC=Senior Companions; ADRD=Alzheimer’s disease and related dementia; SCP= Senior Companion Programs

Abbreviations are not recommended in titles. The title of Figure 1, and all other titles, should be modified. 

ð “Principles of the Senior Companion Programs Plus Training Program (adapted from Merrill, 2002)” instead of ““Principles of the SCP Plus Training Program (adapted from Merrill, 2002)”.

Methods

Participants

-       Lines 120-122: “Additionally, SCs were not allowed to meet with their clients due to risk for infection and the IRB did not allow any in-person data collection.” 

The authors should specify the acronym IRB.

ð Additionally, SCs were not allowed to meet with their clients due to risk for infection and the IRB (Institutional Review Board) did not allow any in-person data 121 collection.”

-       Lines 139-141: The authors wrote: “The training manual used Times New Roman, 22-point font size and as recommended by Czaja et al., 2019, the manual was written in plain, non-technical language.”

This sentence should be replaced with: 

ð “The training manual used Times New Roman, 22-point font size and as recommended by Czaja et al. (2019), the manual was written in plain, non-technical language.”

-       Line 144: The title of the table “Overview of the SCP Plus Training.” should be replaced with:

ð Overview of the Senior Companion Programs Plus Training

The authors should add a legend below the Table specifying the meaning of the acronym SC:

SC=Senior Companions

The presentation of the contents of Table 1 could be improved. Indeed, the reader does not understand why the "contents" are not aligned in the same way.

Data Analysis 

Line 187: The authors have written: “All analyses were conducted with SPSS 28 for Windows. r.” Could they specify “.r.”?

Results

Line 189: Demographics of the SCs

As previously stated, writing acronyms in titles is discouraged. The authors should better write: 

ð Demographics of the Senior Companions

-       Line 190-191: “Table 1 reports the descriptive information of SCs who completed the training (=  11).”  The authors refer to Table 1 but it is Table 2.  The letter N must be written in lower case. The sentence should be replaced with: 

ð “Table 2 reports the descriptive information of SCs who completed the training (= 11).”

Table 2 should be placed in the section called Demographics of the SCs”.

Line 201: “Changes of KAD Scores”. 

As previously stated, writing acronyms in titles is discouraged. The authors should better write:

ð  “Changes of Knowledge of Alzheimer’s disease/dementia scale Scores”. 

Lines 202-206: “Table 2 shows the scores of KAD before and after the SCP Plus. The results showed (bottom line in Table 3) the sum scores of KAD were significantly improved (z = -2.97, p = 0.003) for SC participants, with medium KAD score of 10 (range 5-11) at pre-test, and medium KAD score of 12 at post-test (range 10-14). All of the 11 SC participants showed positive change scores between pre- and post- tests (not shown in Table 2).” 

The authors refer to Table 2 but it is Table 3. The sentence should be replaced with:    

ð “Table 3 shows the scores of KAD before and after the SCP Plus. The results showed (bottom line in Table 3) the sum scores of KAD were significantly improved (z = -2.97, p = 0.003) for SC participants, with medium KAD score of 10 (range 5-11) at pre-test, and medium KAD score of 12 at post-test (range 10-14). All of the 11 SC participants showed positive change scores between pre- and post- tests (not shown in Table 3).”

Line 207: In Table 2, a parenthesis is missing: 

ð divorced 5 (45.5%)

Lines 208-209: “Percentage of SCs in SCP Plus group correctly answering true-false knowledge items about AD (N =11).” 

The title should be replaced with:

ð  Percentage of Senior Companions in Senior Companion Programs Plus group correctly answering true-false knowledge items about AD (n =11).”

In Table 3, in the p value column, row 3, the symbol “§"should be deleted. “N” should be written “n”.

Line 210: “§" This symbol should be deleted. The authors should add the acronyms F and T.

The first line of the Table must not be written in bold or separated from the other lines by a horizontal line.

The p-value should always be in italics. The authors should make the change in the first line.

Lines 224-226 : “When only looking at the KAD items correctly answered after the training, results in Table 2 also showed the overall knowledge of Alzheimer’s disease was relatively high at posttest.” In fact, the authors refer to the Table 3. This sentence should be replaced with:

ð  When only looking at the KAD items correctly answered after the training, results in Table 3 also showed the overall knowledge of Alzheimer’s disease was relatively high at posttest.”

Line 239: The title “Respondence of SCs in SCP Plus group on sense of competence in providing care (N =11).” should be replaced with:

ð Respondence of Senior Companions in Senior Companion Programs Plus group on sense of competence in providing care (n =11).”

Line 240: The authors should replace “N” by “n”.

Table 4: The p-value should be in italics. The authors should make the change in the first line.

The first line of Table 4 should not be separated from the other lines by a horizontal line.

Line 253: The title:” Respondence of SCs in SCP Plus group on preparedness for caregiving (=11).” should be replaced with: 

ð  Respondence of Senior Companions in Senior Companion Programs Plus group on preparedness for caregiving (n =11).”

The authors should replace “N” by “n”.

Table 5: The first line of the Table should not be written in bold or separated from the other lines by a horizontal line.

Discussion

References cited in the Discussion section must have been stated in the Background (e.g., Hancock, et al., 2022; Hughes et al., 2009).

Lines 289-293: “Although research is 289 mixed, focusing on caregiver competency may be an important aspect of multi-component ADRD caregiver interventions (Gaugler, et al., 2021; Liew et al., 2019) and enhancing a sense of mastery (e.g. confidence, competence) is recommended for interventions with African American ADRD caregivers (Bonds Johnson et al., 2022).”

ð e.g.,

Conclusion

Line 337-338: “Interventions that are culturally informed are important, as family caregiving among African American ADRD family caregivers is complex due to the familial structures and the social and historical context of caregiving that are fundamental to many caregivers’ experiences (Brewster et al. 2020).”

A comma is missing in the reference.

ð “Interventions that are culturally informed are important, as family caregiving among African American ADRD family caregivers is complex due to the familial structures and the social and historical context of caregiving that are fundamental to many caregivers’ experiences (Brewster et al., 2020).”

Reviewer 3 Report

In this manuscript the authors dealt with an innovative training approach for Alzheimer’s disease and related dementia. The topic is interesting, and the results of the manuscript may have practical significance.

However, I have a couple of suggestions for authors:

- pay attention to technical errors (following pdf material)

- in the methodology, list the items of the appropriate scales

- state the sample size calculation because the study included a very small number of respondents, which reduces the relevance of the obtained data

- Also, the study included only female subjests. In order to somehow solve the problem, I suggest that the authors add a paragraph in the introduction and/or discussion of the manuscript that would refer to gender differences in dementia. Why only women?

- And finally, a control group of subjects would undoubtedly contribute to the quality of the work: advice to authors to supplement the material with data on control subjects, obtained by subsequent testing of controls.

Round 2

Reviewer 1 Report

Thank you for responding thoroughly to reviewer's comments. I found s a few minor typos throughout (e.g., line 62 has an extra 'a' and 77 has two periods), so I suggest one more careful reader after the edits are accepted.

Author Response

Thank you - we carefully reviewed the manuscript and corrected any typos.

Reviewer 3 Report

The authors took into account all the remarks of the reviewer and thereby improved the quality of the manuscript.

Author Response

Thank you for your time and comments.